# Income inequality and social gradients in children's height: a comparison of cohort studies from five high-income countries

Philippa K Bird ![ORCID],[1,2] Kate E Pickett,[1] Hilary Graham,[1] Tomas Faresjö,[3] Vincent W V Jaddoe,[4,5] Johnny Ludvigsson,[6] Hein Raat,[7] Louise Seguin,[8] Anne I Wijtzes,[7] Jennifer J McGrath[9]

► Additional material is published online only. To view please visit the journal online (http://dx.doi.org/10.1136/bmjpo-2019-000568).

For numbered affiliations see end of article.

**Correspondence to**
Dr Philippa K Bird; philippa.bird@nhs.net

## ABSTRACT

**Background** Health and well-being are better, on average, in countries that are more equal, but less is known about how this benefit is distributed across society. Height is a widely used, objective indicator of child health and predictor of lifelong well-being. We compared the level and slope of social gradients in children's height in high-income countries with different levels of income inequality, in order to investigate whether children growing up in all socioeconomic circumstances are healthier in more equal countries.

**Methods** We conducted a coordinated analysis of data from five cohort studies from countries selected to represent different levels of income inequality (the USA, UK, Australia, the Netherlands and Sweden). We used standardised methods to compare social gradients in children's height at age 4–6 years, by parent education status and household income. We used linear regression models and predicted height for children with the same age, sex and socioeconomic circumstances in each cohort.

**Results** The total analytic sample was 37 063 children aged 4–6 years. Gradients by parent education and household income varied between cohorts and outcomes. After adjusting for differences in age and sex, children in more equal countries (Sweden, the Netherlands) were taller at all levels of parent education and household income than children in less equal countries (USA, UK and Australia), with the greatest between-country differences among children with less educated parents and lowest household incomes.

**Conclusions** The study provides preliminary evidence that children across society do better in more equal countries, with greatest benefit among children from the most disadvantaged socioeconomic groups.

## INTRODUCTION

On average, countries that are more equal have better health and well-being.[1 2] Comparisons of population data have shown that high-income countries with less income inequality have longer life expectancy, lower teenage pregnancy rates, lower infant mortality and better child well-being.[3–8] Within countries, the social gradient in health is well-established: each

### What is known about the subject

► Countries with lower income inequality have better average health and well-being.
► There is a social gradient in child health within countries.
► There is some limited and inconsistent evidence that health is worse at all points of the social gradient in more unequal countries.

### What this study adds

► There is considerable variation in the social gap in child height between countries, suggesting that inequalities are not inevitable.
► Children are shorter at all points of the social gradient in more unequal societies, with the greatest detriment among children from the most disadvantaged backgrounds.

incremental improvement in socioeconomic circumstances brings an associated health gain.[9]

It remains unclear, however, how the health benefit of living in a more equal country is distributed across society—in other words, whether or not both rich and poor do better. One previous ecological study using aggregated mortality data for rich and poor counties within US states found that mortality rates for some causes of death were lower for everyone in more equal US states, but for causes of death with no social gradient, there was no inequality effect.[10] Several studies have used individual data to compare social gradients in health in different countries.[11–17] When they are interpreted in relation to income inequality, they provide some indication that health is better among people at all points in the social hierarchy in more equal countries. For example, a comparative study showed that infant mortality

rates were higher overall and the social gradient was steeper in (less equal) England and Wales than in (more equal) Sweden. Infant mortality was therefore higher in all social classes in England and Wales than in Sweden, with the greatest difference among lower social class groups.[11] However, there are inconsistencies between studies.

There is also ongoing discussion on the relationship between income inequality and health. Several recent reviews have concluded that health and well-being are better in more equal countries.[2 18] There are some differences by study design and outcome, with findings from longitudinal studies less consistent than cross-sectional studies, and some mixed findings from studies on child health.[19 20] The causal mechanisms of the relationship are not fully understood.[21] Investigating how the relationship between income inequality and health varies in relation to socioeconomic position can help to clarify the nature of this relationship.

Understanding whether people in all socioeconomic circumstances do better in more equal countries might offer important potential policy levers to address inequality. When considering *child* health, and health-related indicators of lifelong health and well-being that can be measured in children, the policy implications are significant. If children's trajectories of health and well-being are shaped by both family socioeconomic circumstances and societal levels of inequality, then promoting more equitable health and life chances is a matter of social justice.

Child height is a well-recognised marker of children's social and environmental conditions and general population health. Economic, nutritional and health constraints early in life, as well as psychological and social stressors such as family conflict, reduce children's likelihood of achieving their genetic height potential.[22] Height not only reflects past and current socioeconomic conditions, but is also a good indicator of future well-being and success, including better health and workplace success and higher average income.[23–26] Within countries, numerous studies have described social gradients in growth and height from birth, through childhood and into adulthood.[27 28] However, if children from affluent families are shorter in more unequal countries than equally well-off families in more equal countries, then the need for policies aimed at inequality reduction (as well as poverty alleviation) might be brought into focus.

This study aimed to answer the question: how do social gradients in child height vary in relation to income inequality in high-income countries? We used child height as a marker of current and future health and well-being to investigate whether children growing up in all socioeconomic circumstances do better in more equal countries.

## METHODS
### Study design and participants
We conducted a coordinated analysis of five cohort studies from countries with different levels of income inequality, using identical statistical methods and comparable variables.[29] We compared social gradients in height at age 4–6 years, by household socioeconomic position.

We included three national cohorts—the US Early Child Longitudinal Study K cohort (ECLS-K), the UK Millennium Cohort Study (MCS) and the Longitudinal Study of Australian Children K cohort (LSAC-K); and two regional or city cohorts—the Generation R Study (GenerationR) from Rotterdam in the Netherlands, and All Babies in Southeast Sweden (ABIS) from Southeast Sweden. At the time of data collection, the USA had the highest income inequality (Gini coefficient=37.0), followed by the UK (34.8), Australia (31.1), the Netherlands (26.8), and Sweden (22.2) (table 1).

The cohorts recruited pregnant women or infants, except the US and Australian cohorts, which recruited children at kindergarten entry. As analysis used secondary

**Table 1** Cohort data sets and income inequality

|  | ECLS-K | MCS | LSAC-K | GenerationR | ABIS |
|---|---|---|---|---|---|
| Country (region) | USA | UK | Australia | The Netherlands (Rotterdam) | Sweden (Southeast) |
| Gini coefficient (for year of data analysed) * | 37.0 (**most unequal**) | 34.8 | 31.1 | 26.8 | 22.2 (**most equal**) |
| Year (for data analysed) | 1999 | 2006 | 2004 | 2008–2010 | 2003 |
| First year of data collection | 1998/9 | 2001/2 | 2004 | 2001–2005 | 1997–1999 |
| Child age at recruitment | 4–6 years | 9 months | 4–5 years | During pregnancy | During pregnancy |
| Child age at sweep of data analysed | 4–6 years | 4–6 years | 4–5 years | 4–6 years | 4–6 years |
| Cohort sample size at age 4–6 years | 21 409 | 15 460 | 4983 | 6175 | 7445 |

Source:[48–52]

*Gini coefficients (net) from the Standardized World Income Inequality Database. The data from regional cohorts have been assigned the Gini coefficient for the whole country for this analysis. Comparable data on the Gini coefficients of regions were not available.

ABIS, All Babies in Southeast Sweden; ECLS-K, Early Child Longitudinal Study K cohort; GenerationR, Generation R Study; LSAC-K, Longitudinal Study of Australian Children K cohort; MCS, Millennium Cohort Study.

data only, no additional patient and participant involvement was conducted.

## Processes

There were differences in recruitment and sample between cohorts; therefore, we defined inclusion and exclusion criteria to maximise comparability of samples. Children were excluded if they were born outside the sample country/region, or if they were multiple births. Children were also excluded if they were from minority ethnic groups in the country, consistent with previous cross-national studies,[13] and because average height often varies by ethnicity. The final samples were singleton children aged 4–6 from the majority ethnic group who were born in the country/region in which the cohort took place.

Height was selected as an objective and comparable indicator of child physical health across cohorts. Parent education level and household income were selected as indicators of socioeconomic position due to their wide use in epidemiological research,[30] availability and comparability. To harmonise parent education, we defined four levels using the highest qualification of either parent. In the USA, UK and Australian cohorts these levels were: 1—no secondary school; 2—secondary school; 3—post-secondary/technical; 4—university degree or higher. In the Netherlands and Sweden parents were highly educated, so using the same categories would not capture differences in education status within these samples. We defined education categories for these cohorts which, though technically different, were chosen to have a similar meaning in terms of social status and employment opportunities as the study countries. In the Dutch and Swedish cohorts the levels were: 1—secondary school; 2—lower technical/vocational; 3—higher technical/theoretical; 4—university degree or higher (full explanations of categories are provided in the online supplementary file).

To harmonise household income, we calculated equivalised household income in purchasing power parity dollars (PPP$) at 2005 prices. We first converted household income bands to continuous values using interval regression (used to model outcomes in ordered categories where the exact value of the observation is unknown).[31] In the US cohort we truncated the highest household incomes to improve comparability (as top income bands were lower in other cohorts). We accounted for differences in price and currency using the Organisation for Economic Co-operation and Development (OECD) consumer price index figures to calculate incomes at 2005 prices, then conversion to PPP$ using the OECD PPP$ exchange rates for 2005. Finally, we took account of differences in household size by equivalising income using the square root of the number of people in the household.

## Statistical analysis

We analysed each data set separately, then compared findings. As rates of missing data were generally low (0.2%–2.2%), we conducted complete case analysis. All analyses were conducted using STATA V.11.

Descriptive statistics were calculated for socioeconomic position, health outcomes and other socio-demographic variables. We conducted preliminary unadjusted analysis of child height by parent education level and by quintiles of household income for comparability between cohorts. We then used linear regression to take differences in age and sex into account. We regressed child height on parent education level/household income, child sex and age. We investigated the presence of different relationships by child sex using interactions between socioeconomic exposure variables and sex, and non-linear relationships with income by including a squared income term when significant. Separate models were run for each cohort and for parent education and household income (log transformed for analysis).

Finally, we predicted height from each of the models for children in the same circumstances in each cohort, using the margins command in STATA. For parent education, height was predicted for girls and boys separately, at exactly 5 years of age, with each of the four levels of parent education. For household income, height was predicted for girls and boys, aged exactly 5 years, at the 5th, 25th, 50th, 75th and 95th percentiles of equivalised household income. These were then presented graphically to compare the slope and level of gradients between countries.

We took account of sampling and attrition in the analysis and calculation of standard errors wherever possible. In the USA, UK and Australian cohorts, this was achieved using the svy commands in STATA. In the Dutch and Swedish cohorts, analyses were unweighted as weighting variables were not available. We present model coefficients with 95% confidence intervals; for inclusion of second-order terms in the models (interaction and squared terms), we used a cut-off at p<0.1.

## RESULTS

The total analytic sample was 37 063 children from five cohort studies (table 2). The mean age ranged from 57.0 months in the Australian sample to 72.6 months in the Dutch sample and the male/female proportion was similar across cohorts. Mean height varied considerably, in accordance with the age distribution across cohorts, and was highest in the Dutch cohort. Boys were significantly taller in all cohorts. Parent education levels were highest in the Dutch and Swedish cohorts and considerably lower in the UK cohort; household income was considerably higher in the US cohort than in other cohorts. Unadjusted social gradients in child height are reported in the supplementary file.

Parent education was a significant predictor of child height in adjusted regression analysis in all cohorts (table 3). Gradients in child height by parent education were steepest in the cohorts from the most unequal countries (USA and UK), with marked, but less steep gradients

**Table 2** Child and household characteristics, by cohort

| | ECLS-K (USA) | MCS (UK) | LSAC-K (Australia) | GenerationR (The Netherlands) | ABIS (Sweden) |
|---|---|---|---|---|---|
| Analytic sample size | 9495 | 12523 | 4243 | 3632 | 7170 |
| Child age, months, mean (SD) | 68.8 (4.3) | 62.6 (2.9) | 57.0 (2.6) | 72.6 (3.3) | 64.5 (3.5) |
| Sex, n girls (%) | 4606 (48.0%) | 6097 (48.8%) | 2082 (48.7%) | 1837 (50.6%) | 3408 (47.5%) |
| Height, cm, mean (SD) | 113.6 (5.4) | 110.6 (4.9) | 108.5 (4.7) | 118.9 (5.2) | 114.1 (5.2) |
| Highest parent education level, n (%) | | | | | |
| Level 1 (lowest) | 132 (3.8%) | 1791 (13.7%) | 280 (8.1%) | 168 (4.8%) | 144 (2.0%) |
| Level 2 | 250 (23.5%) | 5259 (42.3%) | 1042 (25.6%) | 741 (21.1%) | 1344 (18.9%) |
| Level 3 | 633 (36.9%) | 1362 (11.1%) | 1374 (34.4%) | 908 (25.8%) | 2246 (31.6%) |
| Level 4 (highest) | 597 (35.9%) | 3936 (32.9%) | 1527 (31.9%) | 1701 (48.4%) | 3371 (47.5%) |
| Household income, 2005 PPP$* | | | | | |
| Mean (SD) | 69570 (48,309) | 41823 (26,163) | 44958 (22,182) | 46172 (17,092) | 43216 (16,196) |
| Median | 58620 | 36428 | 43540 | 44281 | 41954 |

Note: n are unweighted; % are weighted in all cohorts except GenerationR and ABIS.
*measured before tax in ECLS-K; measured after tax in MCS, LSAC-K, GenerationR and ABIS. Converted to PPP$ at 2005 prices.
ABIS, All Babies in Southeast Sweden; ECLS-K, Early Child Longitudinal Study K cohort; GenerationR, Generation R Study; LSAC-K, Longitudinal Study of Australian Children K cohort; MCS, Millennium Cohort Study; PPP$, purchasing power parity dollars.

in the cohorts from Australia and the more equal countries (the Netherlands, Sweden). In the Swedish cohort, the gradient was almost flat except for the lowest education category (which contained only 2% of children). There was no significant interaction between sex and parent education. Figure 1 shows predicted gradients in child height from each model for girls and boys aged exactly 5 years, by parent education level.

Equivalised household income was also a significant predictor of child height in all cohorts (table 4). Gradients were steepest in the cohorts from the three most unequal countries (USA, UK, Australia), and less steep in the cohorts from the most equal countries (Sweden and the Netherlands). There was no significant interaction between sex and household income. Figure 2 shows predicted gradients in child height from each model for

**Table 3** Multivariable regression models of child height, parent education level, child age and sex

| | ECLS-K (USA, Gini=37.0) | MCS (UK, Gini=34.8) | LSAC-K (Australia, Gini=31.1) | GenerationR (The Netherlands, Gini=26.8) | ABIS (Sweden, Gini=22.2) |
|---|---|---|---|---|---|
| **Parent education** | | | | | |
| Level 1 | −1.88 (−2.56 to −1.21) | −1.60 (−1.90 to −1.29) | −0.84 (−1.42 to −0.26) | −0.84 (−1.63 to −0.07) | −1.28 (−2.14 to −0.42) |
| Level 2 | −1.00 (−1.26 to −0.74) | −0.99 (−1.24 to −0.73) | −0.46 (−0.82 to −0.11) | −0.44 (−0.87 to −0.02) | −0.12 (−0.43 to 0.20) |
| Level 3 | −0.34 (−0.62 to −0.07) | −0.72 (−1.07 to −0.36) | −0.39 (−0.71 to −0.07) | −0.34 (−0.74 to −0.05) | −0.11 (−0.37 to 0.16) |
| Level 4 (baseline) | 0.00 | 0.00 | 0.00 | 0.00 | 0.00 |
| (Wald test for difference between parent education categories) | p<0.01 | p<0.01 | p<0.01 | p=0.04 | p<0.01 |
| **Age (months)** | 0.47 (0.44 to 0.50) | 0.55 (0.52 to 0.58) | 0.52 (0.46 to 0.57) | 0.58 (0.53 to 0.63) | 0.56 (0.52 to 0.59) |
| **Sex—boy** | 0.83 (0.59 to 1.07) | 0.96 (0.75 to 1.17) | 1.01 (0.73 to 1.29) | 0.63 (0.30 to 0.95) | 0.90 (0.67 to 1.14) |
| **No. observations** | 9282 | 12182 | 4191 | 3512 | 6464 |

ABIS, All Babies in Southeast Sweden; ECLS-K, Early Child Longitudinal Study K cohort; GenerationR, Generation R Study; LSAC-K, Longitudinal Study of Australian Children K cohort; MCS, Millennium Cohort Study.

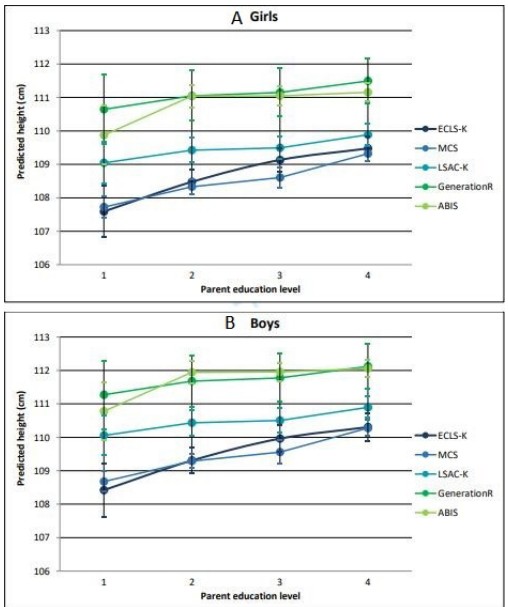

**Figure 1** Predicted gradients in child height for (A) girls and (B) boys aged exactly 5 years, by parent education level.

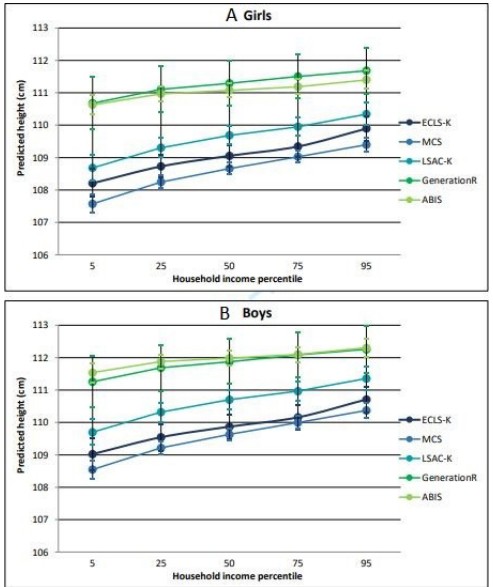

**Figure 2** Predicted gradients in child height for (A) girls and (B) boys aged exactly 5 years, by equivalised household income percentile (p5–p95).

girls and boys aged exactly 5 years, by household income level.

Children were taller in the cohorts from the most equal countries at every parent education or income level, with the greatest difference between cohorts among children with the least educated parents (a 'fanning out' pattern; see figures 1 and 2).

## DISCUSSION

We compared social gradients in child height to explore whether children in all socioeconomic circumstances do better in more equal countries. After adjusting for differences in age and sex, children were shorter on average and the social gradient was steepest in the cohorts from the most unequal countries. A 'fanning out' pattern was evident, with children in more equal countries taller at all levels of parent education, but with the greatest

between-country differences among children with less educated parents. Analysis of social gradients in relation to household income indicated broadly similar patterns.

Findings from comparison of harmonised cohort studies may reflect methodological differences between the studies, errors in the comparative method or actual population differences.[32] We needed to analyse each cohort separately, as data access requirements did not enable pooling of the data sets. The coordinated analysis enabled a high degree of harmonisation, increasing confidence that findings reflect actual population differences. By including only singleton children aged 4–6 from the majority ethnic group, we minimised the extent to which comparative findings can be explained by differences in age or ethnicity of children between the cohort samples. However, as the Swedish and Dutch cohorts had subnational samples, findings may not be representative of the country as a whole. In particular, GenerationR is from a

**Table 4** Multivariable regression models of child height, household income, child age and sex

|  | ECLS-K (USA, Gini=37.0) | MCS (UK, Gini=34.8) | LSAC-K (Australia, Gini=31.1) | GenerationR (The Netherlands, Gini=26.8) | ABIS (Sweden, Gini=22.2) |
|---|---|---|---|---|---|
| **Log equiv. household income (PPP$)** | −0.95 (−1.56 to −0.34) | 0.85 (0.69 to 1.01) | 0.89 (0.63 to 1.15) | 0.71 (0.29 to 1.13) | 0.68 (0.35 to 1.01) |
| **Log equiv. household income (PPP$) ^2** | 0.08 (0.04 to 0.11) | | | | |
| **Age (months)** | 0.47 (0.45 to 0.50) | 0.55 (0.52 to 0.58) | 0.52 (0.46 to 0.58) | 0.58 (0.53 to 0.63) | 0.56 (0.52 to 0.59) |
| **Sex—boy** | 0.82 (0.57 to 1.06) | 0.97 (0.76 to 1.18) | 1.02 (0.73 to 1.31) | 0.58 (0.25 to 0.91) | 0.91 (0.68 to 1.14) |
| **No. observations** | 9257 | 12 170 | 4073 | 3311 | 6455 |

ABIS, All Babies in Southeast Sweden; ECLS-K, Early Child Longitudinal Study K cohort; GenerationR, Generation R Study; LSAC-K, Longitudinal Study of Australian Children K cohort; MCS, Millennium Cohort Study; PPP$, purchasing power parity dollars.

city with relatively high inequality and poverty in relation to other parts of the Netherlands.[33] Some differences in harmonised variables may also have limited comparisons. Parent education categories were chosen to have a similar social meaning, but were technically different between countries. Using identical categories would have resulted in over 95% of children being in the top two categories in the Swedish and Dutch cohorts. Finally, incomes in the UK, Australian, Swedish and Dutch cohorts were measured after tax; however incomes in the Australian cohort were measured before tax (so may be higher and ordered differently than if they were measured after tax).

A relationship between income inequality and height has previously been established. In a recent analysis of data from 169 countries, income inequality was shown to be a greater predictor of average height than absolute national income.[34] Our findings also provide further evidence in support of previous studies showing better health among people at all points in the social hierarchy in more equal countries.[11–16]

To understand the meaning of differences in height between countries and by socioeconomic position, the relative contributions of genetic height potential and environmental conditions need to be unpacked. Although, at an individual level, genetic differences explain the majority of variation in body height,[35] when comparing populations (eg, socioeconomic groups or country populations), environmental factors are thought to play the largest role.[36] Population heights have increased over time, alongside improvements in living standards within countries, and height has often been used as a marker of societal living conditions.[37 38] The Dutch population, for example, is now the tallest in the world after an increase in average height of over 20 cm since the 1850s.[39] A number of international studies have shown little variation between populations and ethnic groups in the growth of infants and children from affluent, educated families or when nutritional and health needs are met.[40–43] This suggests that when environmental conditions are optimal, genetic factors play a limited role in ethnic variations in height.

In this study, living standards in the cohort countries vary, and interpreting the relative contribution of genetic potential and socioeconomic differences is complex. Parental height, an important predictor of child height,[44] reflects this complex interplay between genetic and environmental factors. While including it in the analysis would have adjusted for genetic differences, it also would have adjusted for environmental differences that our research sought to identify. There are also complex patterns and inequalities in relation to ethnicity. In the UK, for example, children from Asian and Black backgrounds are taller than white children, despite living in more deprived areas.[28] We therefore analysed the sample from the majority ethnic group only (white European heritage in all cohorts) to minimise concerns about differences in genetic potential, and ensure that comparisons reflect environmental differences between

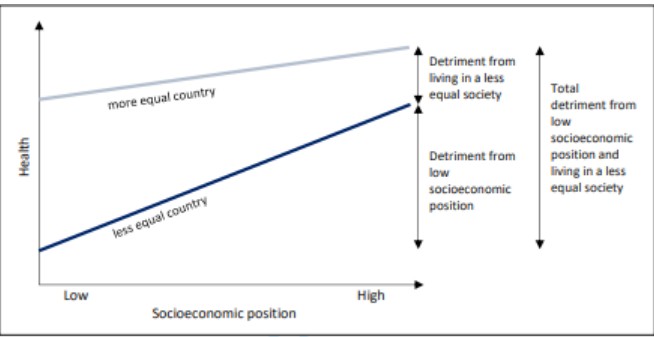

**Figure 3** Layers of child health and development detriment.

socioeconomic groups and countries. There were considerable differences between the cohorts in the country of origin and socioeconomic position of families from ethnic minority backgrounds, as well as health outcomes. Therefore, this approach enabled a clearer focus on the research question, but also resulted in patterns in some population groups not being compared, and may have affected the patterns of inequalities observed. Further research would be useful to identify and compare the extent of inequalities by ethnicity between cohorts from different countries.

There are number of potential explanations for a relationship between income inequality and the social gradient in child health. Living in an unequal country may have psychosocial effects on children and their families. Stressors of family life related to living in an unequal country, such as increased mental illness, child maltreatment and long working hours, increased status differentiation and lower social cohesion, are all likely to affect health.[1] More equal countries also often have more generous welfare systems and more equitable public infrastructure, which may influence both population health and health inequalities.[45 46] Other contextual differences between countries, such as gross national income, may play a role, although in work not reported here we did not find a clear relation with social gradients in the study countries. Future research using different data sets and outcomes would enable further exploration of this relationship.

In the context of growing income inequality in high-income countries, it is important to understand the implications of income inequality for population health.[47] First, we have demonstrated cross-national variation in the social gap in child height. This suggests that such differences are not inevitable and could be avoided through appropriate policies and interventions. These inequalities in child height are likely to have long-term implications for health and well-being later in life. Second, this analysis provides evidence that children across society do better in more equal societies; although the greatest benefit is among children from less advantaged backgrounds, even children within the most advantaged families may do better in more equal societies. Our findings suggest a picture of multiple, overlapping and interacting socioeconomic contexts (figure 3). Children

in less advantaged socioeconomic circumstances in unequal societies may experience a double detriment—from individual socioeconomic circumstances and from living in an unequal country.

**Author affiliations**
[1]Department of Health Sciences, University of York, York, UK
[2]Leeds Teaching Hospitals NHS Trust, Leeds, UK
[3]Department of Medical and Health Sciences, Linköping University, Linköping, Sweden
[4]Generation R Study Group, Erasmus Medical Center, Rotterdam, The Netherlands
[5]Department of Pediatrics, Erasmus Medical Center, Rotterdam, The Netherlands
[6]Division of Pediatrics, Medical Faculty, Linköping University, Linköping, Sweden
[7]Department of Public Health, Erasmus Medical Center, Rotterdam, The Netherlands
[8]Department of Social and Preventive Medicine, Universite de Montreal, Montreal, Québec, Canada
[9]Department of Psychology, Concordia University, Montreal, Québec, Canada

**Acknowledgements** We would like to thank Jonathan Bradshaw and members of the International Network for Research on Inequalities in Child Health (INRICH) for their advice and support for the study.

**Contributors** PKB contributed to the study design, conducted the data analysis and literature search and drafted the report. KEP and HG contributed to the study design, interpretation of findings and writing of the report. TF, VWVJ, JL, HR, LS, AIW and JJM contributed to access to the cohort data, harmonisation of data, interpretation of findings and writing the report. All authors edited the report and approved the final draft.

**Funding** This work was primarily supported by a doctoral training studentship from the UK Economic and Social Research Council. Further support was provided by grants awarded by the Canadian Institutes of Health Research (#MSH95353, #MOP123533, #00309MOP-123079). This project received funding from the European Union's Horizon 2020 research and innovation programme under grant agreement 733206 (LifeCycle Project). The Early Child Longitudinal Study K cohort is managed by the National Center for Education Statistics, US Department of Education. For the Millennium Cohort Study, we are grateful to the Centre for Longitudinal Studies, Institute of Education for the use of these data and to the UK Data Archive and Economic and Social Data Service for making them available. However, they bear no responsibility for the analysis or interpretation of these data. The Longitudinal Study of Australian Children was initiated and is funded by the Australian Government Department of Families, Housing, Community Services and Indigenous Affairs, and is undertaken in partnership with the Australian Institute of Family Studies and the Australian Bureau of Statistics. The Generation R Study (GenerationR) is conducted by the Erasmus Medical Center, Rotterdam in close collaboration with the School of Law and Faculty of Social Sciences of the Erasmus University Rotterdam, the Municipal Health Service Rotterdam area, the Rotterdam Homecare Foundation and the Stichting Trombosedienst en Artsenlaboratorium Rijnmond, Rotterdam. We gratefully acknowledge the contribution of general practitioners, hospitals, midwives and pharmacies in Rotterdam. The first phase of the GenerationR was made possible by financial support from the Erasmus Medical Center, Rotterdam, the Erasmus University Rotterdam and the Netherlands Organisation for Health Research and Development. We are grateful to the All Babies in Southeast Sweden (ABIS) team at Linkoping University, all families, well-baby clinics and schools. ABIS has been funded by Swedish Research Council (No. K2009-70X-21086-01-3), grants from The Swedish Council for Working Life and Social Research (No. 2008-0284), Medical Research Council of Southeast Sweden, Swedish Child Diabetes Foundation (Barndiabetesfonden), Juvenile Diabetes Research Foundation and Research Council for Southeast Sweden.

**Disclaimer** The funders of the study had no role in the design of the study, collection, analysis and interpretation of data, or in writing this manuscript.

**Competing interests** TF, VWVJ, JL, HR, LS, AIW and JJM report no conflicts of interest. PKB reports funding from an Economic and Social Research Council (ESRC) doctoral studentship and the Canadian Institute for Health Research (CIHR). HG reports funding from an ESRC doctoral studentship. KEP reports grants from the ESRC during the conduct of the study, and is co-founder and trustee of The Equality Trust, which campaigns for greater income equality.

**Patient consent for publication** Not required.

**Ethics approval** The study involved analysis of secondary data only. All cohort studies had been reviewed and approved by appropriate ethics review boards and obtained informed consent from participants.

**Provenance and peer review** Not commissioned; externally peer reviewed.

**Data availability statement** Data from the MCS, ECLS and LSAC are available in a public, open access repository. Data from GenerationR and ABIS are available on request (https://generationr.nl; http://www.abis-studien.se). .

**ORCID iD**
Philippa K Bird http://orcid.org/0000-0002-9601-7979

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
