## [Reviewer comments · BMJ Paediatrics Open]

ARTICLE DETAILS

TITLE (PROVISIONAL)	Income inequality and social gradients in children's height: a comparison of cohort studies from five high-income countries
AUTHORS	Bird, Philippa; Pickett, Kate; Graham, Hilary; Faresjö, Tomas; Jaddoe, Vincent; Ludvigsson, Johnny; Raat, Hein; Seguin, Louise; Wijtzes, Anne; McGrath, Jennifer

VERSION 1 – REVIEW

REVIEWER	Reviewer name: Peter Flom Institution and Country: Peter Flom Consulting USA Competing interests: None
REVIEW RETURNED	22-Aug-2019

GENERAL COMMENTS	I confine my remarks to statistical aspects of this paper. The general approach is fine, but I have some issues that need to be resolved before I can recommend publication. General: What is a "coordinated analysis"? What about parent height? That surely varies by country and is related to child height p 5 line 54 or so (BMJ uses an odd line numbering thta doesn't match the text) what are NLSCY and QLSCD? p 6 line 4 What are lower tech and higher tech? line 9 Please describe interval regression a little as it will likely be unfamiliar to most readers (I think that term is mostly used by Stata and that "interval censored regression" may be more usual). line 25 Please describe "social gradients". line 26 Why use quintiles? Table 2 For income, give median and IQR instead of (or in addition to) mean and sd
---

REVIEWER	Reviewer name: Richard Layte Institution and Country: Trinity College Dublin Competing interests: None
REVIEW RETURNED	23-Sep-2019

GENERAL COMMENTS	There is now a veritable library of research examining the "income inequality hypothesis" (IIH) which asserts that above a critical threshold of national income, life expectancy depends more on the distribution of income than the absolute level. This paper seeks to contribute to the empirical analysis of this proposition using another empirical measure of health outcomes: child height.
---

	As the paper makes clear, child height is a good barometer of overall health as it's a well-known marker of environmental conditions and general population health. The paper makes a useful contribution to the area by using micro-data from five child cohort studies which, crucially, have widely divergent levels of income inequality (US, UK, Australia, Netherlands & Sweden). Whilst I think this is a good paper with interesting findings, I would suggest that the authors give more of an indication of the variability in the empirical literature in the area. At the moment, the literature review suggests that the IIH is now well established and that there is no doubt about the empirical generalisation. In fact, there are plenty of important papers using longitudinal methods such as country or regional fixed effects which have found no relationship between income inequality and health (e.g Beckfield 2004; Adjaye-Gbewonyo et al. 2018; Neumayer and Plümper 2016; Avendano 2012; Hill and Jorgenson 2018; Hu, van Lenthe, and Mackenbach 2015; Leigh and Jencks 2007; Mellor and Milyo 2001, 2003; Modrek and Ahern 2011; Torre and Myrskylä 2014). My sense would be that the paper would be strengthened by showing that there is some doubt about the underlying hypothesis else, why show new evidence? Second, I think the general approach that the authors take to analysis is broadly right here but have concerns about the tests used. Ideally the paper would draw upon a statistical sample of countries which vary in terms of their level of income inequality but given the limitation to five, a country effects approach is appropriate. Here analyses are carried out on individual countries rather than pooling the observations and employing country dummies. This means that comparisons of absolute and relative difference between countries become rather problematic, particularly for income, as they rely on the confidence intervals around the means. A better approach would be to pool the observations and use interactions between country dummies and education/income group in models stratified by sex as tests of difference can then be employed using the margins command in STATA. It would be useful if the authors included more discussion of the consequences of removing the ethnic minority groups from their samples for the comparisons between countries. The composition of minority groups vary across samples and there is a strong association between minority status and SES which is likely to vary by country. Removing minorities may have important consequences for the pattern of inequalities observed.
--	--

REVIEWER	Reviewer name: Ana Paula Muraro Institution and Country: Public health institute, Federal University of Mato Grosso, Brazil Competing interests: None
REVIEW RETURNED	27-Sep-2019

GENERAL COMMENTS	Reviewer's report: Clearly, the paper deals with an important theme, with good data from cohort studies. The manuscript has the potential to make a useful contribution to the literature. However, I consider that the article needs some few adjustments. Please, find below my suggestion: In the introduction section:  - Second paragraph: the authors cite references 11 to 17 on studies that have used individual data to compare social gradients in height, but some of these references has not evaluated height.
--

	The methods section is well described, but I have some doubt: - I am confused if children from Canada were evaluated (as said in Abstract), once table 1 do not mention this country. Results: In general, mainly findings were clearly exposed, but I suggest that the authors show predicted gradients in child height for boys, maybe in the same "figure", using (A) and (B). Discussion: In general, mainly findings were clearly exposed, the hypotheses was presented, and the organization of the text in this section was great.
--	---

VERSION 1 – AUTHOR RESPONSE

Reviewer: 1

I confine my remarks to statistical aspects of this paper. The general approach is fine, but I have some issues that need to be resolved before I can recommend publication.

General: What is a "coordinated analysis"?

Response: We have clarified in the text at the start of the methods on page 4 and added a reference to a paper explaining this approach. "We conducted a coordinated analysis of five cohort studies from countries with different levels of income inequality, using identical statistical methods and comparable variables [29]."

What about parent height? That surely varies by country and is related to child height

Response: This is a good point and it relates to our discussion of the role of genetic differences in height. We have added text to clarify this in the discussion on page 10. "Parental height, an important predictor of child height [49], reflects this complex interplay between genetic and environmental factors. Whilst including it in the analysis would have adjusted for genetic differences, it also would have adjusted for environmental differences that our research sought to identify."

p 5 line 54 or so (BMJ uses an odd line numbering that doesn't match the text) what are NLSCY and QLSCD?

Response: This was an error and these acronyms have been removed from page 5.

p 6 line 4 What are lower tech and higher tech?

Response: As different qualifications and terminology are used in each country, it would be difficult to explain these categories in a succinct way in the methods. We have added a reference to the supplementary file, where full definitions of the categories are provided.

line 9 Please describe interval regression a little as it will likely be unfamiliar to most readers (I think that term is mostly used by Stata and that "interval censored regression" may be more usual).

Response: We have added a short explanation in brackets after the term 'interval regression' on page 6: "(used to model outcomes in ordered categories where the exact value of the observation is unknown)".

line 25 Please describe "social gradients".

Response: We have removed this term from the sentence, as it was unnecessary and made it a little unclear. The sentence now reads: "We conducted preliminary unadjusted analysis of child height by parent education level and by quintiles of household income".

line 26 Why use quintiles?

Response: quintiles of income were used to enhance comparability between cohorts (the original income bands were not comparable). This is now clarified in the text.

Table 2 For income, give median and IQR instead of (or in addition to) mean and sd

Response: We have added the median income to Table 2. Unfortunately, we did not run the IQR on income in the original analysis and no longer have access to some of the datasets to be able to run this.

Reviewer: 2

There is now a veritable library of research examining the “income inequality hypothesis” (IIH) which asserts that above a critical threshold of national income, life expectancy depends more on the distribution of income than the absolute level. This paper seeks to contribute to the empirical analysis of this proposition using another empirical measure of health outcomes: child height. As the paper makes clear, child height is a good barometer of overall health as it’s a well-known marker of environmental conditions and general population health. The paper makes a useful contribution to the area by using micro-data from five child cohort studies which, crucially, have widely divergent levels of income inequality (US, UK, Australia, Netherlands & Sweden).

Whilst I think this is a good paper with interesting findings, I would suggest that the authors give more of an indication of the variability in the empirical literature in the area. At the moment, the literature review suggests that the IIH is now well established and that there is no doubt about the empirical generalisation. In fact, there are plenty of important papers using longitudinal methods such as country or regional fixed effects which have found no relationship between income inequality and health (e.g Beckfield 2004; Adjaye-Gbewonyo et al. 2018; Neumayer and Plümper 2016; Avendano 2012; Hill and Jorgenson 2018; Hu, van Lenthe, and Mackenbach 2015; Leigh and Jencks 2007; Mellor and Milyo 2001, 2003; Modrek and Ahern 2011; Torre and Myrskylä 2014). My sense would be that the paper would be strengthened by showing that there is some doubt about the underlying hypothesis else, why show new evidence?

Response: This is an interesting point. We have added a paragraph in the introduction to explore the inconsistencies in the literature and explain how this adds further justification for the paper (page)

“There is also ongoing discussion on the relationship between income inequality and health. Several recent reviews have concluded that health and wellbeing are better in more equal countries.[2,18] There are some differences by study design and outcome, with findings from longitudinal studies less consistent than cross sectional studies, and some mixed findings from studies on child health.[19,20] The causal mechanisms of the relationship are not fully understood.[21] Investigating how the relationship between income inequality and health varies in relation to socioeconomic position can help to clarify the nature of this relationship.”

In addition, our paper adds new evidence on the differential experience of living more/less equal countries, by socioeconomic position (much of the previous literature has focussed on average health).

Reviewer: Second, I think the general approach that the authors take to analysis is broadly right here but have concerns about the tests used. Ideally the paper would draw upon a statistical sample of countries which vary in terms of their level of income inequality but given the limitation to five, a country effects approach is appropriate. Here analyses are carried out on individual countries rather than pooling the observations and employing country dummies.

This means that comparisons of absolute and relative difference between countries become rather problematic, particularly for income, as they rely on the confidence intervals around the means. A better approach would be to pool the observations and use interactions between country dummies and education/income group in models stratified by sex as tests of difference can then be employed using the margins command in STATA.

Response: We agree with the reviewer that pooling the data would have allowed some very interesting analysis. Our analysis was limited by access to datasets – some of the datasets had to be accessed in the country of origin, and unfortunately it was not possible to pool them. Therefore we chose to use a coordinated analysis in order to ensure that analyses were conducted in the most similar way possible and then compare results.

We have explained this in the discussion (p9): “We needed to analyse each cohort separately, as data access requirements did not enable pooling of the datasets. The coordinated analysis enabled a high degree of harmonisation, increasing confidence that findings reflect actual population differences.”

It would be useful if the authors included more discussion of the consequences of removing the ethnic minority groups from their samples for the comparisons between countries. The composition of minority groups vary across samples and there is a strong association between minority status and SES which is likely to vary by country. Removing minorities may have important consequences for the pattern of inequalities observed.

Response: This is an important point. This approach was justified to ensure that the research question was answered, but will have had consequences for the patterns seen – and it is important to acknowledge this. We have added the following on page 10:

“There were considerable differences between the cohorts in the country of origin and socioeconomic position of families from ethnic minority backgrounds, as well as health outcomes. Therefore, this approach enabled a clearer focus on the research question, but also resulted in patterns in some population groups not being compared, and this may have affected the patterns of inequalities observed. Further research would be useful to identify and compare the extent of inequalities by ethnicity between cohorts from different countries.”

Reviewer: 3

Reviewer's report:

Clearly, the paper deals with an important theme, with good data from cohort studies. The manuscript has the potential to make a useful contribution to the literature. However, I consider that the article needs some few adjustments. Please, find below my suggestion:

In the introduction section:

- Second paragraph: the authors cite references 11 to 17 on studies that have used individual data to compare social gradients in height, but some of these references has not evaluated height.

Response: Thank you for pointing this out – it should have read “social gradients health”. This has now been corrected: “Several studies have used individual data to compare social gradients in health in different countries”

The methods section is well described, but I have some doubt:

- I am confused if children from Canada were evaluated (as said in Abstract), once table 1 do not mention this country.

Response: Children from Canada were not evaluated for this paper – we have removed “Canada” from the abstract.

Results: In general, mainly findings were clearly exposed, but I suggest that the authors show predicted gradients in child height for boys, maybe in the same “figure”, using (A) and (B).

Response: These have been added

Discussion: In general, mainly findings were clearly exposed, the hypotheses was presented, and the organization of the text in this section was great.